# Time-of-flight detection of terahertz phonon-polariton

Tianchuang Luo [1,3], Batyr Ilyas [1,3], A. von Hoegen [1,3], Youjin Lee[2], Jaena Park[2], Je-Geun Park [2] & Nuh Gedik [1] ✉

A polariton is a fundamental quasiparticle that arises from strong light-matter interaction and as such has attracted wide scientific and practical interest. When light is strongly coupled to the crystal lattice, it gives rise to phonon-polaritons (PPs), which have been proven useful in the dynamical manipulation of quantum materials and the advancement of terahertz technologies. Yet, current detection and characterization methods of polaritons are still limited. Traditional techniques such as Raman or transient grating either rely on fine-tuning of external parameters or complex phase extraction techniques. To overcome these inherent limitations, we propose and demonstrate a technique based on a time-of-flight measurement of PPs. We resonantly launch broadband PPs with intense terahertz fields and measure the time-of-flight of each spectral component with time-resolved second harmonic generation. The time-of-flight information, combined with the PP attenuation, enables us to resolve the real and imaginary parts of the PP dispersion relation. We demonstrate this technique in the van der Waals magnets $NiI_2$ and $MnPS_3$ and reveal a hidden magnon-phonon interaction. We believe that this approach will unlock new opportunities for studying polaritons across diverse material systems and enhance our understanding of strong light-matter interaction.

Exposing crystals to a strong external electromagnetic (EM) field can induce highly nontrivial dynamics[1-4]. Under such experimental conditions, a new type of quasiparticle, a polariton, can emerge[5,6]. Polaritons are hybrid light-matter modes that arise from strong light-matter interaction and therefore carry crucial information about the interaction strength and hold promising potential in applications such as high-speed, low-loss communication[7,8]. Polaritons can be of various types, such as phonon-polaritons (PPs)[9], magnon-polaritons[10-12], exciton-polaritons[13,14] and others[6], depending on the specific nature of the light-matter interaction. In particular, PPs arise when light is strongly coupled to the crystal lattice and its collective excitations, i.e. phonons. This is schematically illustrated in Fig. 1a. When an EM wave propagates inside a crystal, it can induce a coherent oscillation of infrared-active lattice vibrations if they fall within the EM field spectral bandwidth. The interaction with this dipolar vibration influences the dispersive properties of the medium. The dipolar motion of the

infrared-active vibration leads to re-radiation of the EM wave, which propagates and in-turn re-excites the dipolar vibration, giving rise to propagating hybrid-light-lattice modes, i.e. PPs. These modes have been actively explored as a platform for THz waveguides[15,16] and as a tool to study nonlinear phononic effects and dynamically control material properties[17-19].

The physics of PPs is described by the dispersion relation (Fig. 1b). The real part of the PP dispersion relation features an avoided crossing (Fig. 1b, color-coded line), the magnitude of which provides information about their coupling strength. Correspondingly, the imaginary part of the dispersion (Fig. 1b, blue line), which describes the damping of the PP with propagation, exhibits a resonance peak around the phonon frequency, indicating fast dissipation of the electromagnetic field energy through the phonon mode. However, determining the PP dispersion relation is a highly nontrivial task. The challenge lies in identifying the wavevector (momentum) of each spectral component

[1]Department of Physics, Massachusetts Institute of Technology, Cambridge 02139 MA, USA. [2]Department of Physics and Astronomy, Seoul National University, Seoul, South Korea. [3]These authors contributed equally: Tianchuang Luo, Batyr Ilyas, A. von Hoegen. ✉e-mail: gedik@mit.edu

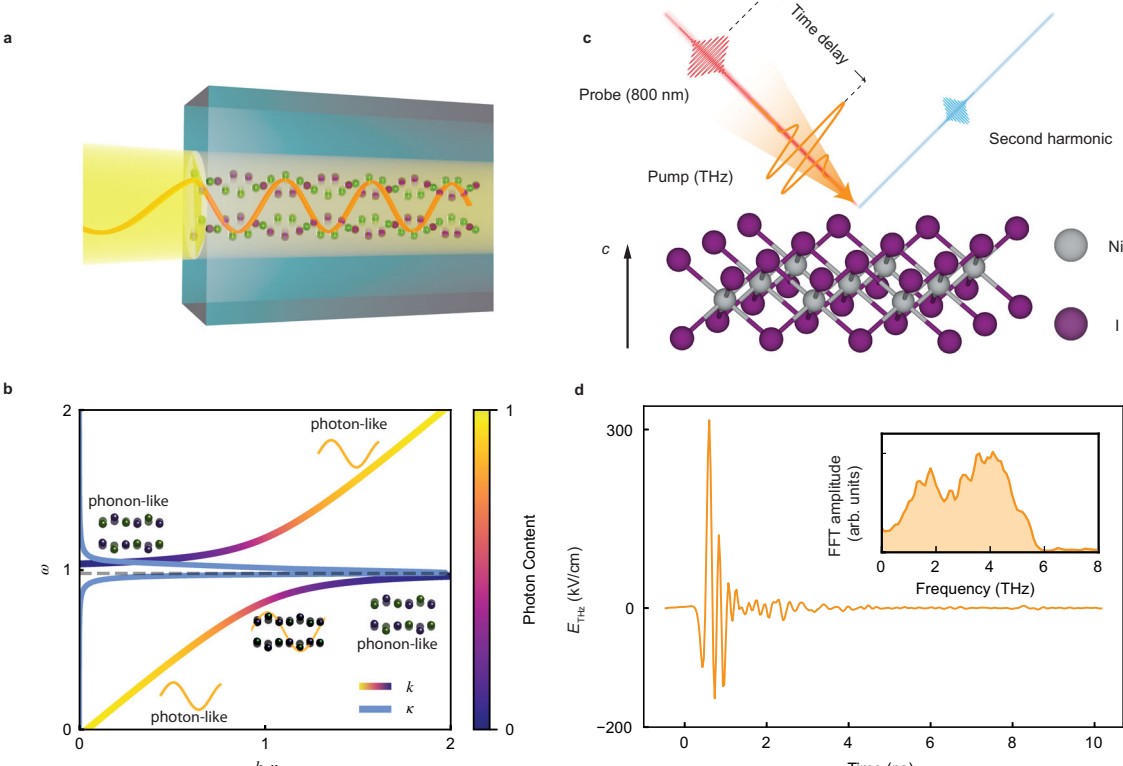

**Fig. 1 | Experimental setup and illustration of PP. a** Schematics of the PP. Free space electromagnetic wave (the orange curve on the left) hits the sample (the blue cube) and induces phonon oscillations (green and purple spheres) with nonzero dipole moment. Phonon and the light wave propagate together and form PP. **b** A typical dispersion relation of PP with schematics illustrating the photon and phonon components in different regions of the dispersion relation. The real part ($k$) of the dispersion curve is color-coded by the photon weight. The imaginary part ($\kappa$) of the dispersion is shown in blue. **c** Schematics of the experimental setup. Intense THz pump (orange) and the 800 nm probe (red) pulses are incident on the *ab* surface of the NiI$_2$ sample at 45°. The second harmonic of the 800 nm probe reflected from the *ab* surface (blue) is collected as the probe signal. **d** The profile of the terahertz field. Inset: the spectrum of the terahertz, which covers the range of 0–6 THz.

of the PPs. The small wavevector range (1 mm$^{-1}$–1 µm$^{-1}$) where photon and phonon momenta cross renders optical techniques to be the most sensitive to PP dispersion relations. Traditional methods such as Raman scattering[20,21] and transient grating spectroscopy[18,22] require careful tuning of the incidence angle of the photon to determine the wavevector of the polariton. In addition, these methods are only applicable to inversion-symmetry broken systems where the polaritons are simultaneously Raman active. Other methods extract the polariton wavevector by satisfying the phase-matching condition[23,24], where a frequency sweep is necessary to map out the dispersion. Recent advances in the generation of stable terahertz (THz) laser pulses have opened new possibilities to directly measure the polariton dispersion by using time-domain THz spectroscopy (TDTS)[25]. However, this method determines the PP wavevector by measuring the exact phase delay of the THz EM wave inside the sample, which requires electro-optic sampling in an external crystal and careful phase referencing. In addition, free space TDTS is only suitable for the study of macroscopic samples due to the large diffraction limit of the THz EM wave. Moreover, these methods have difficulties in obtaining the phonon oscillation amplitude carried by the PPs, which is an important piece of information to study lattice nonlinearities.

Here, we propose a method to measure the PP dispersion based on the time-of-flight of coherent PPs. In this method, we resonantly launch PPs within the wide bandwidth of an ultrashort THz pulse, and measure the time required for each frequency component to travel through the sample's bulk to directly obtain the real part of the dispersion relation. The idea behind this approach is the equivalence of the dispersion relation and the group velocity as a function of

frequency:

$$v_{\mathrm{g}}(\omega) = \frac{\mathrm{d}\omega}{\mathrm{d}k}. \qquad (1)$$

A similar time-of-flight technique has been used to measure the dispersion of near-infrared (NIR) exciton polaritons[26]. By combining this idea with state-of-the-art THz techniques, we demonstrate a time-of-flight study of the PP dispersion in van der Waals (vdW) layered magnetic materials. The amplitude and phase-resolved detection is realized with time-resolved optical second harmonic (SH) generation (SHG) of a NIR probe pulse. The reflected SHG light intensity as a function of the time delay between the THz excitation and NIR probe clocks the arrival time of the PPs at the surface (see Fig. 1c). This approach is capable of mapping out the PP dispersion relation without the need of any external adjustment parameter, phase referencing, or external electro-optic sampling. The imaginary part of the dispersion can also be obtained by comparing the SHG amplitude of the PP before and after traversing the sample. Finally, in contrast to conventional TDTS techniques, detection at NIR wavelengths enables us to extend this approach to measure microscopic samples below the THz diffraction limit.

## Results

As a proof-of-concept, we apply this time-of-flight method to the layered multiferroic material NiI$_2$. Below 59 K NiI$_2$ develops a multiferroic ground state, which can persist down to atomic monolayer limit[27]. Static magnetoelectric responses of NiI$_2$ have been studied[28],

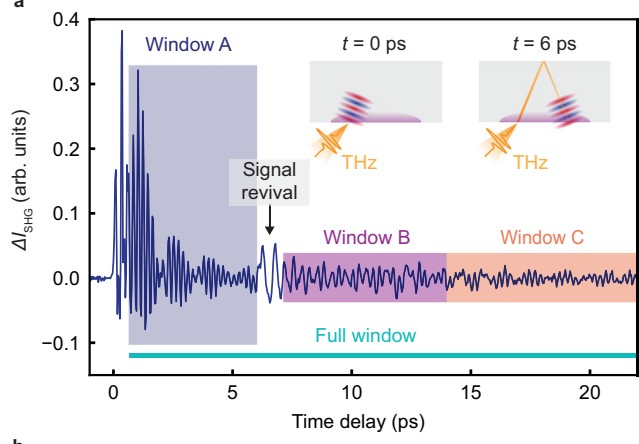

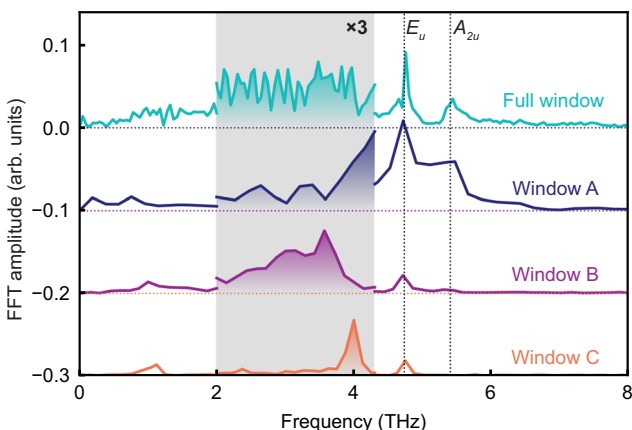

**Fig. 2 | THz field induced SHG in NiI₂. a** THz field induced SHG in NiI₂ as a function of delay time between THz pump and NIR probe pulses. The inset shows the schematics of the dominant contribution to the signal before and after the signal revival. The gray rectangle is the sample. The orange pulse is the THz pump, and the purple-shaded region is the sensing area of the SHG probe. The periodic wave in red and blue is the excited PP. The cyan horizontal line and the shaded regions color-code different time windows used to perform FFT. **b** FFT of the pump-probe time trace within different time windows. Two vertical dashed lines indicate the frequencies of the $E_u$ mode (4.7 THz) and the $A_{2u}$ mode (5.3 THz). Three-time magnification is applied in the shaded region for clarity.

however its dynamical counterpart, the coupling between magnons and phonons, has not been reported. Understanding such interactions is important for identifying effective pathways for the control of macroscopic properties, with potential applications such as high-speed storage devices. The broken inversion symmetry of the NiI₂ multiferroic ground state leads to a finite $\chi^{(2)}$ optical nonlinearity, which results in a static SHG from the NIR probe pulses. Any coherently oscillating infrared-active mode will dynamically modulate $\chi^{(2)}$ and will therefore be visible as changes in the SHG intensity as a function of time delay $\Delta I_{SHG}(t) \sim \Delta \chi^{(2)}(t)$ at the frequencies of those modes (see Supplementary Note 2).

We use optical rectification of 1300 nm laser pulses in an organic nonlinear crystal to generate THz pulses that cover the low energy phonon modes in NiI₂ (see Methods). The peak-to-peak THz electric field at the sample position is 350 kV/cm and the spectral weight is centered around 3 THz with a bandwidth of 6 THz (Fig. 1d).

Figure 2a shows the THz-field-induced change of the SHG intensity of NiI₂ at 7 K, deep in the multiferroic state. Near zero time delay, when the THz pump and the NIR probe pulses are temporally over-lapped, we observe THz-field induced SHG (TFISH, see Supplementary Note 2 for more details). At later times, we observe coherent

oscillations with multiple frequencies, resulting in a beating pattern (Fig. 2a, Window A). These oscillations can be ascribed to infrared-active collective modes of the lattice (phonons) and the spins (magnons)[27,29]. Notably, following the initial decay of the oscillations, we observe a sudden revival of the oscillation amplitude at around 6 ps time delay.

The coherent oscillations and the revival can be understood in the following way. NiI₂ is opaque to the SH of our NIR probe pulses ($\hbar\omega_{SH} = 3.1$ eV) and therefore the reflected SHG intensity is sensitive only to changes of $\chi^{(2)}$ close to the sample surface within the pene-tration depth of the SH light[30,31]. Therefore, the initial coherent oscil-lations can be understood as non-propagating collective modes in the system, which are probed close to the sample's surface (Fig. 2a, left inset). The revival of the oscillations at 6 ps can not be reconciled with non-propagating modes. Replicas of pump or probe pulses outside the sample can also be ruled out since such revival is not observed in our electro-optic sampling traces (see Fig. 1d). Therefore, such a signal suggests a propagating wavepacket of PPs that got reflected from the sample's back surface and reached the front surface after around 6 ps (Fig. 2a, right inset). We note that since the penetration depth of SH is smaller than the THz wavelength, the phase-matching condition is unimportant for the detection of PP in our case.

We reveal unconventional dynamics of the coherent oscillations by obtaining the Fourier spectrum of the time trace in Fig. 2a. Fig-ure 2b shows the fast Fourier transform (FFT) with a time window that excludes the initial TFISH signal (labeled as Full window). The result (Fig. 2b, shown in cyan) shows two sharp peaks at 4.7 THz and 5.4 THz, which have been previously assigned to infrared-active $E_u$ and $A_{2u}$ modes in the high-temperature phase, respectively[29]. In addition, we observe a dense set of peaks between 1 THz and 4.5 THz (Fig. 2b, shaded region), in stark contrast to the phonon modes and is reminiscent of interference of two time-delayed wavepackets. In this case, we expect different Fourier spectra from different time windows. To test this hypothesis, we performed FFT analysis for different time windows (see Fig. 2a, b). The Fourier spectrum of time window A (before the oscillation revival) shows a clear phonon spectrum with two peaks. These peaks are broadened due to the shorter time window. As described above, this observation agrees well with non-propagating phonon modes. For window B (after the signal revival), we additionally observe a broad peak between 2 and 4 THz, with a lineshape very different from that of the phonons. At larger time delays, this feature sharpens and moves to higher fre-quencies (window C).

Next, we perform a complete time-frequency analysis utilizing a wavelet transform (see Methods for technical details) of the time-dependent SH trace to better understand the spectral weight between 2 and 4 THz (see Fig. 3a). The wavelet transform convolves the time trace with a sliding wavelet at different scales and better resolves the time-frequency relation compared to short-time FFT (STFT) (see Supplementary Note 3). This analysis yields the instantaneous spec-trum for each delay in the SH time trace. Close to zero delay, we observe our excitation spectrum and its second harmonic frequencies (see Supplementary Note 2). The frequencies of the $E_u$ and $A_{2u}$ modes appear as horizontal lines along the time axis. After $t = 6$ ps, coin-cidental with the revival of the coherent oscillations in Fig. 2a, a chirped feature emerges. Its frequency increases with time, starting below 2 THz and gradually saturates to around 4 THz at larger delays. This chirped feature corresponds to the peak between 2 THz and 4 THz in the FFT of window B and C (Fig. 2b) and does not depend on the strength of the THz driving field (Supplementary Note 4), establishing it as the linear response of the system. We can therefore assign this chirped mode to the spread-out PP wavepacket that was subject to its dispersion relation.

To further establish this conclusion, we perform a comprehensive finite-difference-time-domain (FDTD) simulation of the THz optical

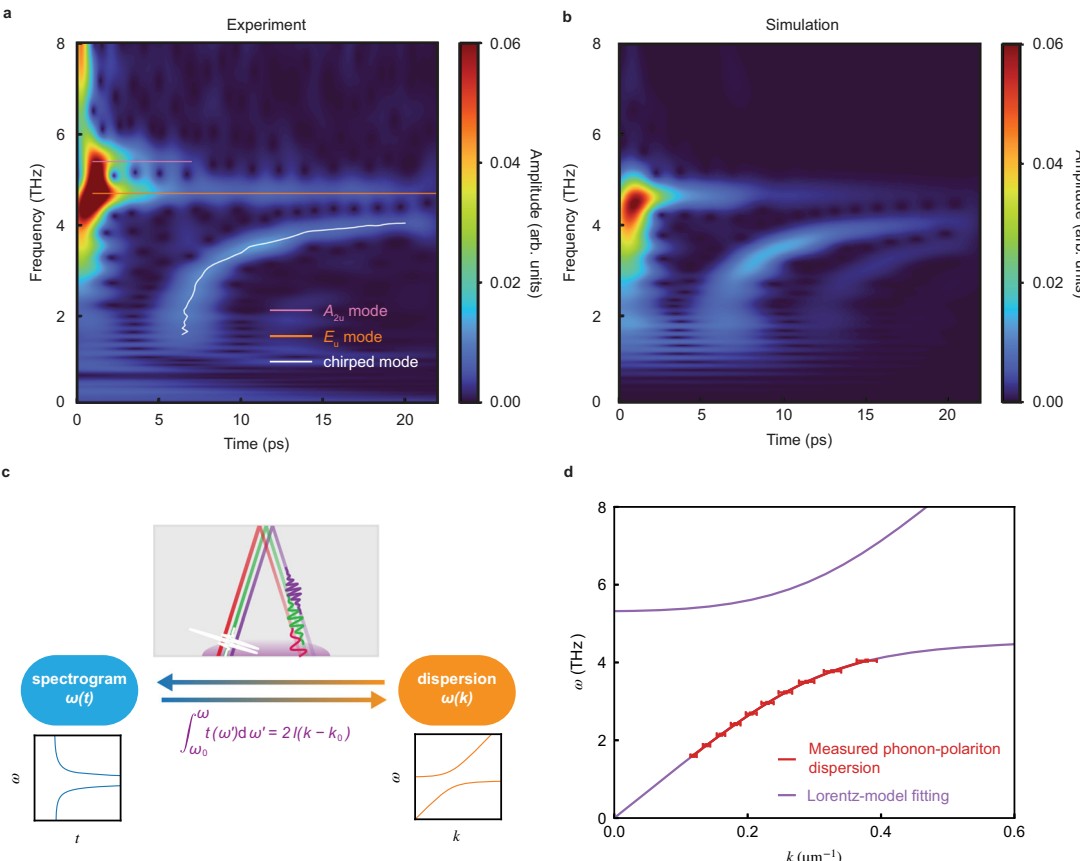

**Fig. 3 | Obtaining the real part of the PP dispersion relation from the spectrogram. a** Spectrogram of the THz pump SHG probe time trace in Fig. 2a. White line is the extracted peaks of the chirped mode. Orange and pink lines are guides to the eye. **b** Spectrogram of the simulated polarization at the sample surface after the THz pump. The simulation takes into account of the electric polarization due to the $E_u$ mode. Both the $E_u$ mode and the chirped mode in the experimental spectrogram are well captured. **c** Relation between the PP dispersion $\omega(k)$ and the spectrogram $\omega(t)$. Upper panel: schematics of the physical picture of the chirped mode. The ultrashort broadband THz pump pulse is converted to a PP wavepacket inside NiI$_2$

(white). Different frequency components travel with different group velocities to the back surface of the sample and reflects back to the front (red, green, and purple pulses). Smaller frequency components travel faster. The shaded purple region is the region SHG probe can sense. **d**. Reconstructed PP dispersion from the spectrogram in (**a**). The red curve is obtained by integrating Eq. (2) with experimentally determined $\omega(t)$ and the violet curve is a Lorentz model fitting. The error bars show the standard error of wavevector and frequency of our constructed dispersion relation at representative points.

response of NiI$_2$. To capture the essence of the experiment, we model the system as a THz plane wave incident perpendicularly to the sample surface. This approximation is valid since, given the THz refractive index of NiI$_2$ ($n \gtrsim 3$)[29], the angle between the THz wavevector inside the sample and the sample normal has only a small effect (see Supplementary Note 5 for discussions). We input the profile of our THz pulse (see Fig. 1d) and combine Maxwell's equations with the dynamical response of the $E_u$ phonon mode, which can be modeled by a damped harmonic oscillator (see Methods for simulation details). The electric polarization at the sample surface as a function of time is used to model our experimental SHG data. The result of these simulations is shown in Fig. 3b and we find a similar chirped response as in the experiment, confirming that the chirped response indeed arises from a propagating PP wavepacket, and that the $E_u$ phonon mode is responsible for the formation of the observed PP.

Therefore, we conclude that the chirped mode originates from the propagating wavepackets of the PP (shown in the schematics in Fig. 3c). As the free space strong THz field hits the sample surface, it resonantly excites a set of PP wavepackets which propagate in the medium. As the polariton frequency approaches the phonon resonance it becomes increasingly phonon-like and its group velocity decreases. As a result, the time each frequency components of the PP (red, green and purple lines in Fig. 3c) needs to travel to the back surface and then return to the

front surface of the sample is different and is captured by the frequency–delay time relation of the chirped mode.

We use this result to extract the travel time of the different frequency components $t(\omega)$ of the polariton wavepacket, from which the dispersion relation of the PP can be directly deduced (Fig. 3c). The time ($t$) and frequency ($\omega$) information of the PP feature is extracted by Gaussian fitting of linecuts of the spectrogram, which yields a time uncertainty of 0.05 ps and frequency uncertainty of 0.03 THz (see Supplementary Note 6). This time–frequency resolution is comparable to that achieved by time-domain THz spectroscopy. The group velocity of the PP $v_g(\omega)$ can be straightforwardly extracted as $v_g(\omega) = \frac{2l}{t(\omega)}$. Here $2l$ denotes the optical path length, which is approximated by twice of the sample thickness, with an error of less than 3% (see Supplementary Note 5). The dispersion relation $\omega(k)$ is related to the group velocity through $v_g(\omega) = \frac{d\omega}{dk}$ and we can obtain it by integrating the above relation:

$$\int_{\omega_0}^{\omega} t(\omega')d\omega' = 2l(k - k_0). \tag{2}$$

Therefore, the dispersion relation is reconstructed with only one undetermined additive constant $k_0$, which can be determined by fitting to polariton dispersion models or requiring the extrapolation of the

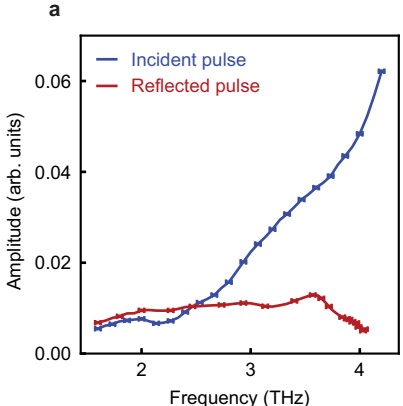
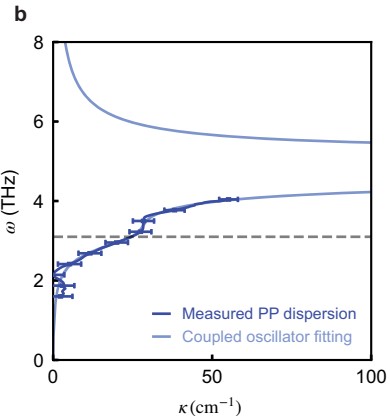
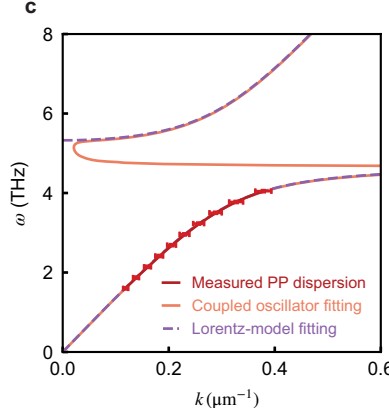

**Fig. 4 | Obtaining the imaginary part of the PP dispersion relation from the spectrogram. a** The amplitude of the incident pulse (blue curve) and the reflected pulse (red curve) as a function of frequency extracted from the spectrogram in Fig. 3a. The error bars show the standard error of the amplitude and frequency at representative points. **b** The experimentally measured imaginary part of the dispersion relation obtained from Eq. (3) (dark blue curve) and its fitting with coupled Lorentz oscillator model (light blue curve). **c** The red curve shows the real part of the dispersion relation as in Fig. 3d. The dashed violet curve is fitting with Lorentz model without damping as in Fig. 3d. The orange curve is the fitting with bi-linear coupling model. The error bars in (**b**) and (**c**) show the standard error of wavevector and frequency of our constructed dispersion relation at representative points.

dispersion relation to satisfy certain constraints. The reconstructed PP dispersion relation of $NiI_2$ is shown in Fig. 3d together with a fit using a Lorentz model without damping (see Methods for more details). The uncertainty of the derived dispersion relation is shown by the error-bars at representative points, which is discussed in detail in Supplementary Note 6. The averaged frequency error is 0.03 THz and wavevector error is 9 mm$^{-1}$. In addition, we note that the same analysis can be applied if we probe the transmitted PP at the back side of the sample[32]. However, the crucial information on the PP attenuation will be lost, as discussed below.

Besides the real part of the dispersion relation, the attenuation of the PP with propagation length can also be extracted from our time-of-flight method, which is related to the imaginary part ($\kappa$) of the dispersion ($\tilde{k} = k + i\kappa$). We can extract the PP oscillation amplitude as the amplitude of the spectrogram in Fig. 3a within the incident pulse $A_0$ (Fig. 4a blue line) and the reflected pulse $A_1$ (Fig. 4a red line) at different frequencies. They are related to each other by the imaginary part of the wavevector $\kappa$ and the reflection coefficient $r$ at the boundary by

$$\frac{A_1}{A_0} = e^{-2\kappa l} r(1+r) S, \tag{3}$$

where the factor $(1+r)$ accounts for the contribution of both the reflected pulse and its further reflection at the front surface. $S$ is an order-one constant correction factor that accounts for the slight spatial offset between the incident pulse and the reflected pulse (see Methods). The frequency dependence of $\kappa$, as obtained from Eq. (3), is shown in Fig. 4b. The divergence of $\kappa$ near the phonon resonance frequency is observed, which is consistent with the damped harmonic oscillator model (see Methods). Notably, a shoulder-like peak in $\kappa$ is observed around $\omega_2 = 3.1$ THz. We rule out the possibility of this peak originating from data analysis or our experimental configuration (see Supplementary Note 5). Such anomalous damping behavior is due to coupling of the $E_u$ mode to another mode, which opens additional energy dissipation pathway to the coupled mode[33,34] (see Methods). Since $\kappa(\omega)$ does not change with the THz fluence (see Supplementary Note 7), we conclude that this is a bi-linear coupling between the $E_u$ mode to another mode. The frequency of the shoulder-like peak in $\kappa$ is close to a magnon mode in $NiI_2$[27], indicating that our observation is a potential signature of magnon-phonon coupling. The real and the imaginary parts of the dispersion can be simultaneously fitted with the bi-linear coupling model[34] (Fig. 4b, c). The coupling strength between

the two modes is $\frac{b_{12}b_{21}}{(2\pi)^4} = (1.7 \pm 0.1)$ THz$^4$, while the $E_u$ mode oscillator strength $\frac{b_{1E}b_{E1}}{(2\pi)^2\varepsilon_0} = (61 \pm 2)$ THz$^2$ (see Methods). The observation of the magnon-phonon coupling is due to the type-II multiferroic nature of $NiI_2$, and hints towards controlling the magnetization across the sample with PP.

In addition, the amplitude of the PP can also be measured by comparing with the static SHG signal from $NiI_2$. The result is shown in the Supplementary Note 8. We can separate the contributions from electronic and phonon polarizations using the parameters from the fitting in Fig. 3d. The peak of the phonon component is about 0.2% of the static SHG signal in the low-temperature multiferroic phase.

While in $NiI_2$ only the lower branch of the PP is visible due to finite THz bandwidth, in principle, our time-of-flight approach is sensitive to both the lower and upper PP branches (see Supplementary Note 10). To demonstrate the versatility of this time-of-flight approach, we carried out a similar measurement on a different non-centrosymmetric vdW magnet $MnPS_3$[35]. In this experiment, we not only observe the dispersively chirped mode, but also a second replica at larger time delays, which originates from multiple reflections of the PP in the sample (see Fig. S10). Notably, the upper branch of the PP is also visible. The reconstructed PP dispersion relation of $MnPS_3$ is presented in Supplementary Note 11. The light-matter coupling strength is $\frac{b_{1E}b_{E1}}{4\pi^2\varepsilon_0} = (50.6 \pm 0.4)$ THz$^2$, indicating a weaker coupling to in-plane polarized light in the 1–5 THz range as compared to $NiI_2$.

## Discussion

Compared with other methods for studying PPs, our time-of-flight approach offers several unique advantages. With a single scan, a broad bandwidth of phonon polariton dispersion can be obtained, as long as its attenuation coefficient is below 100 cm$^{-1}$ for typical samples. This eliminates the need for repeated measurements with varying incidence angles or wavelengths. Larger attenuation coefficients usually only appear close to phonon resonances, where the PP is predominantly composed of non-propagating evanescent waves. The accessible range of the polariton dispersion can potentially be further extended by the use of broader bandwidth THz pulse. While time-domain THz spectroscopies have been used to investigate PPs, they require additional alignments to sample the THz pulse in an electro-optic sampling crystal. In contrast, our method enables sampling the time-dependent, phase-resolved PP amplitude directly at the sample

surface. This aspect also allows the study of PPs over microscopic length scales, smaller than the THz diffraction limit. Since we are decoding the back-reflected THz PP in both time–frequency space, we are less affected by the overlapping back-reflected response that can limit the frequency resolution of a time-domain THz spectroscopy. Finally, anharmonic PP effects can potentially be observed in crystals with large nonlinearities, provided with our high THz field strength.

In summary, we demonstrated a time-resolved technique to measure both the real and the imaginary parts of coherent PP dispersion. This time-of-flight method is based on broadband THz excitation of a PP wavepacket and its detection using time-resolved SHG, which offers several technical advantages over traditional methods. Utilizing linear or nonlinear probing in the NIR or visible spectrum, as demonstrated here, shifts the achievable spatial resolution down to the ~1 μm range far beyond what conventional TDTS can achieve. Our time-of-flight approach can also be extended to other polaritonic modes, such as magnon polariton. We expect that this method will be highly useful for the study of low-energy collective modes and their interactions with light. It may also help enable the use of large amplitude PP oscillations as a tool for ultrafast control of material properties.

## Methods

### Sample preparation
Single crystal $NiI_2$ was synthesized via a chemical vapor transport method. Nickel powder (99.99% Sigma-Aldrich) and iodine (99.99%, Alfa Aesar) were mixed in the stoichiometric ratio and sealed in an evacuated quartz tube. The quartz tube was placed in a two-zone furnace with a temperature configuration of 750 °C (hot zone) and 720 °C (cold zone) over 7 days. We used energy-dispersive X-ray spectroscopy and powder X-ray diffraction to check the synthesized crystals' quality and crystallinity. The thickness of the $NiI_2$ under study is $(250 \pm 20)$ μm as characterized by a Bruker Dektak DXT-A stylus profilometer.

Single crystal samples of $MnPS_3$ were synthesized by a chemical vapor transport method. We put a quartz ampoule containing the raw materials into a horizontal two-zone furnace with a temperature configuration of 780 °C (hot zone) and 730 °C (cold zone). The sample thickness is $(85 \pm 10)$ μm as characterized by the profilometer.

### THz pump, SHG probe spectroscopy
The broadband THz pump is obtained by pumping N-benzyl-2-methyl-4-nitroaniline (BNA) crystal with 1300 nm output from an optical parametric amplifier at a repetition rate of 1 kHz. An 800 nm pulse from a Ti-Sapphire amplifier co-propagating with the THz pump pulse is focused on the sample to probe the lattice and polariton dynamics. The incidence angle of the THz and 800 nm pulse is 45°. The THz pump is s-polarized, with its electric field along the in-plane direction. The probe is polarized to maximize the static SHG signal. The SHG from the 800 nm pulse is separated from the fundamental beam by a pair of dichroic mirrors and a narrow band bandpass filter and then directed to a photomultiplier tube (PMT). The output of PMT is collected by a data acquisition (DAQ) card. A schematics of the experimental setup is shown in Supplementary Note 1.

### Wavelet transform
The wavelet transform is performed with the generalized Morse wavelet with $\beta = 120$, $\gamma = 3$[36]. The wavelet transform better resolves the time–frequency relation compared to short-time Fourier transform, which is demonstrated in Supplementary Note 3.

### FDTD simulation
The PP simulations were performed with the FDTD method[37], using an open-source software package[38]. Figure 3b shows the result of the simulation in one spatial dimension. A more realistic simulation with focused THz beam and oblique incidence produces similar results and is available in Supplementary Note 5.

The sample is modeled by a Lorentz polarization model with a single resonance at the $E_u$ phonon mode frequency, where we treat the IR-active phonon mode as a harmonic oscillator with certain damping rate:

$$\left(\frac{\partial^2}{\partial t^2} + \Gamma \frac{\partial}{\partial t} + \omega_T^2\right)P_L = \varepsilon_0 \varepsilon_{r\infty}(\omega_L^2 - \omega_T^2)E, \quad (4)$$

where $\omega_T$ and $\omega_L$ are the transverse and the longitudinal phonon frequency of the $E_u$ mode. $\Gamma$ is the damping rate of the phonon polariton. $\varepsilon_0$ is the vacuum permeability constant and $\varepsilon_{r\infty}$ is the relative permeability at the high frequency limit. The parameters $\omega_T$, $\omega_L$, and $\varepsilon_\infty$ are taken from ref. 29. Decay rate $\Gamma$ is set to 0.04 THz to approximate the experimentally measured decay rate. The total electric polarization of the sample is given by

$$P = \varepsilon_0(\varepsilon_{r\infty} - 1)E + P_L = \varepsilon_0(\varepsilon_{r\infty} + \varepsilon_{rL} - 1)E, \quad (5)$$

where $P_L$ is the phonon contribution and $\varepsilon_0(\varepsilon_{r\infty} - 1)E$ is the electronic contribution.

The THz pump is modeled by a plane wave incident from the sample surface with the same time trace as measured by electro-optic sampling (Fig. 1d). The total electric polarization $P$ at the sample front surface is collected as a function of time, whose spectrogram is shown in Fig. 3b.

The FDTD simulation is performed with a grid size of 1 μm and time step of 2 fs.

### Lorentz model fitting of the PP dispersion relation
After the numerical integration given by Eq. (2), a dispersion curve is obtained. We first fit the integrated dispersion curve with theoretical PP dispersion without damping as the observed small damping has a negligible effect on the real part of the dispersion:

$$\omega(k)^2 = \frac{1}{2\varepsilon_{r\infty}}\left(c^2k^2 + \varepsilon_{r\infty}\omega_L^2 \pm \sqrt{(c^2k^2 + \varepsilon_{r\infty}\omega_L^2)^2 - 4\varepsilon_{r\infty}c^2k^2\omega_T^2}\right), \quad (6)$$

where $c$ is the speed of light. The fitted values are $\omega_T = 4.68$ THz, $\omega_L = 5.27$ THz, which are in good agreement with previous FTIR results in ref. 29.

To include damping, it is more intuitive to consider the following differential equations[39]:

$$\begin{aligned}\ddot{Q}_1 + \Gamma_1\dot{Q}_1 &= -\omega_T^2 Q_1 + b_{1E}E, \\ P &= b_{E1}Q_1 + (\varepsilon_{r\infty} - 1)\varepsilon_0 E,\end{aligned} \quad (7)$$

where $Q$ is the phonon mode coordinate, $b_{1E}$ and $b_{E1}$ are coupling coefficients, which are related to $\omega_L$ by $b_{1E}b_{E1} = \varepsilon_{r\infty}\varepsilon_0(\omega_L^2 - \omega_T^2)$. By coupling Eq. (7) to Maxwell's equations, the dispersion relation is obtained as

$$k + i\kappa = \frac{\omega}{c}\sqrt{\varepsilon_{r\infty} - \frac{b_{1E}b_{E1}/\varepsilon_0}{\omega^2 - \omega_T^2 + i\omega\Gamma_1}}. \quad (8)$$

Based on Eq. (7), coupling to other modes can be straightforwardly implemented by adding another coordinate $Q_2$[34]:

$$\begin{aligned}\ddot{Q}_1 + \Gamma_1\dot{Q}_1 &= -\omega_T^2 Q_1 + b_{1E}E + b_{12}Q_2, \\ \ddot{Q}_2 + \Gamma_2\dot{Q}_2 &= -\omega_2^2 Q_2 + b_{21}Q_1, \\ P &= b_{E1}Q_1 + (\varepsilon_{r\infty} - 1)\varepsilon_0 E.\end{aligned} \quad (9)$$

$b_{12}$ and $b_{21}$ are coupling constants between mode $Q_1$ and $Q_2$. The terms involving $b_{2E}$ and $b_{E2}$ are omitted since we do not observe SH modulation at frequency $\omega_2$. The PP dispersion in this case is

$$k + i\kappa = \frac{\omega}{c}\sqrt{\varepsilon_{r\infty} - \frac{b_{1E}b_{E1}/\varepsilon_0}{\omega^2 - \omega_T^2 + i\omega\Gamma_1 - \frac{b_{12}b_{21}}{\omega^2 - \omega_2^2 + i\omega\Gamma_2}}}. \tag{10}$$

The fitting with Eq. (10) is presented in Fig. 4b, c, which gives $\frac{\Gamma_1}{2\pi} = (0.05 \pm 0.01)$ THz, $\frac{\Gamma_2}{2\pi} = (1.04 \pm 0.04)$ THz, and correction factor $S = 1.28 \pm 0.01$. The fitted values of $\frac{b_{1E}b_{E1}}{\varepsilon_0}$, $b_{12}b_{21}$, and $\omega_2$ are presented in the main text.

## Data availability

Source data are provided with this paper. Other datasets generated and/or analyzed during the current study are available from the corresponding author upon request.

## Code availability

The code used for FDTD simulations during the current study is available from the corresponding author upon request.

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

## Acknowledgements

We acknowledge support from the US Department of Energy, BES DMSE (data taking and analysis), and Gordon and Betty Moore Foundation's EPiQS Initiative grant GBMF9459 (instrumentation) (T.L., B.I., A.v.H., and N.G.). A.v.H. gratefully acknowledges funding from the Alexander von Humboldt Foundation. Work at SNU was supported by the Leading

Researcher Program of the National Research Foundation of Korea (grant number 2020R1A3B2079375) (Y.L., J.P., and J.-G.P.).

## Author contributions

T.L. and B.I. performed the THz experiments. T.L., B.I., and A.v.H. analyzed the experimental data. T.L. performed the FDTD simulations with help from A.v.H. Y.L., J.P., and J.-G.P. synthesized and characterized the $NiI_2$ and $MnPS_3$ crystals. T.L., B.I., A.v.H., and N.G. wrote the manuscript with crucial input from all other authors. This project was supervised by N.G.

## Competing interests

The authors declare no competing interests.
