## [Peer Review File · Nature Communications]

Time-of-flight detection of terahertz phonon-polaritonREVIEWER COMMENTS

Reviewer #1 (Remarks to the Author):

The authors have developed and demonstrated a method for the measurement of dispersion of terahertz phonon polariton by measuring a time of flight of an infrared probing beam.

As stated in the manuscript, dispersion of phonon polaritons has been characterized already almost 20 years ago using time-domain THz spectroscopy. On the one hand, the authors claim that a "careful phase referencing" is not needed (like in THz electro-optic sampling). On the other hand, the stabilized phase shape of the THz pulses is also implicitly required by the reported method. Then, both methods actually rely on a precise measurement and control of the delay between two ultrashort optical pulses.

The data analysis involves a wavelet transform which inevitably leads to uncertainties in time and frequency. It would be helpful if the authors commented on the accuracy and precision of the presented method, and ideally compared it with that achieved using time-domain THz spectroscopy.

The sample thickness must be specified (it is crucial not only for quantitative analysis of the dispersion relation, but also for the correct interpretation of the time-dependence of the probe signal). The temperature of the measurement has to be specified in the manuscript to clarify which phase is actually studied.

The work is certainly relevant and interesting, but I doubt that it is suitable for Nature Communications. I'm convinced that after appropriate revisions, the manuscript would be perfectly suitable for a journal like Phys. Rev. B. There are numerous interesting and important results, which are now hidden in the Supplementary Information while they would deserve publication in the body of a regular paper.

Questions and technical comments:

- * Why magnetic materials and particularly van der Waals magnets were chosen for the study?
- * The explicit relation between $\omega(k)$ and $v_g(\omega)$ should be presented in Eq. (1)
- * The beams hit the sample at 45 degrees while the simulation is performed in 1D. How was the oblique incidence reflected in the simulations and in the results?
- * Many references are incomplete (mostly a page/article number is missing) [10, 17, 23, 24, 27, ...].
- * The spheres in Fig. 1a look grey rather than purple as written in the caption
- * Line 86 the authors claim that "... we can rule out a replica of our pump or probe pulses (see Fig. 1d)". Why the replicas are ruled out?
- * The discussion in the paragraph at lines 89 - 100 could be improved; it is now difficult to decode the main message.
- * Would it be possible to observe also the upper polariton branch using the developed approach?

Reviewer #2 (Remarks to the Author):

Review Report

Manuscript: #NCOMMS-23-27251

Manuscript Title: "Time-of-flight detection of terahertz phonon polariton", by Tianchuang Luo et al.

The authors propose a new approach based on a time-of-flight measurement to determine the phonon-polariton dispersion of van der Waals magnets NiI₂ and MnPS₃. Although their proposed technique is basically interesting, questions are raised after reading the paper as stated below.

(1) The authors said, "Determining the polariton dispersion relation is a highly nontrivial task, and traditional methods such as Raman scattering and transient grating spectroscopy require careful tuning of the incidence angle of the photon to determine the wavevector of the polariton". This is actually true. However, the conventional Raman scattering method tracks Raman active modes, while the authors' proposed technique can access the IR active modes. Therefore, these might be complementary to each other. Even utilizing the authors' technique, I guess, the phonon-polariton (PP) dispersion of Raman active modes cannot be obtained.

(2) To reveal not only the PP dispersion but also the anharmonicity of the PP is crucial for understanding the underlying physics of quantum materials and advancing THz technologies. In previous papers [Refs. 18, 22, 23, 24 and JAP 123, 174103, 2018], anharmonic phonon-polariton characteristics such as damping rate, coupling and so on, are observed. However, the experimental results in this paper seems within a linear regime. Does the authors' technique evaluate the anharmonicity of the PP? If not, the proposed technique is not powerful so much.

(3) The dispersion of phonons and magnons is generally obtained by using neutron scattering method, which may give us the dispersions in the whole Brillouin zone. In contrast, optical measurements only tracks the dispersion around the gamma point. Could the authors compare the measured PP dispersion of NiI₂ and MnPS₃ with the data obtained by neutron scattering?

(4) The authors should state the experimental condition more clearly.

What is the sample thickness? To implement the authors' time-of-flight measurement, the dataset of each temporal window (Window A, B and C in Fig. 2) is separately needed, implying that overlapping of the dataset among the windows might be undesirable. In addition, as seen in the inset of Fig. 2a and Fig. 1S, the incident angle of THz and probe pulses is tilted against the sample, implying that the optical distance is not the sample thickness. Moreover, I guess, the authors need information on the refractive index of the probe.

The regions of the dispersion are limited around $10 \sim 35k(\text{pi}/l)$ in Fig. 3 and $3 \sim 17k(\text{pi}/l)$ in Fig. S5. What kind of the experimental condition contributes to the limitation of the observable region? Could the authors change the region of the dispersion? In case of the previous works using the phase-match conditions [23, 24, JAP 123, 174103, 2018], the dispersion region can be changed by changing the pump wavelength.

The authors utilized the wavelet analysis. How much does the frequency resolution coming from the analysis?

Is the sample the uniaxial crystal or isotropic one? The polarization of THz and probe pulses to the crystal axis is probably crucial for the analysis.

Overall, the paper is well written and basically interesting. However, the authors should claim more clearly the technical merit of their experimental technique or pros and cons to the previous works.

Reviewer #3 (Remarks to the Author):

The authors report a method to measure phonon polariton dispersion curves with a time of flight method. The front surface of a sample is excited with a broadband THz pulse and phonon polaritons are detected after propagation through the sample.

The technique is interesting and could be useful, but in its current form, the manuscript reads as a brief introduction to a new technique and analysis method. The impact of this manuscript would be greater (and better suited for Nature Communications) with the additional information:

- 1) Besides a measurement of a portion of the phonon-polariton dispersion curve, some discussion of what this means and why it is important for the studied materials is needed. Why are NiI_2 and MnPS_3 potentially useful and what information about the materials and their usefulness does the dispersion curve give us? They introduce the interest in polaritons generally in the intro, but then never tie it together with the experimental results.
- 2) What are the limitations of this experimental method? For what phonon-polariton attenuation coefficient versus the sample thickness is the experiment still possible? (The PP needs to be large enough amplitude to still be detected after propagating across the sample)
- 3) Typically, PP dispersion curves are plotted as two parts - the dispersive part and the absorptive part. Can you extract the absorptive part from this data? If so, please include it in the plots.
- 4) With the emphasis on a time-of-flight type measurement, I naturally want to know for what situations would you need/want to probe from the back side of the sample? And what would be considerations for doing the measurement like that?
- 5) I see fitted values for the Lorentz model fitting of the PP dispersion relation for NiI_2, but they are absent from the SI for MnPS_3

Minor things that need addressing:

- A) The text, letters, in the figures I think is too small
- B) Do we have error bars on the values for the measured PP dispersion? For a proof of principle measurement like this, introducing the technique along with it's limitations is important.

Nil2 Reviewer Report

Response to Reviewer #1

The authors have developed and demonstrated a method for the measurement of dispersion of terahertz phonon polariton by measuring a time of flight of an infrared probing beam.

We thank the reviewer for their careful reading and accurate summary of the core results of our manuscript. As we will lay out in the following, we performed more extensive data analysis and added more discussions to the manuscript based on the specific remarks made by the reviewer.

As stated in the manuscript, dispersion of phonon polaritons has been characterized already almost 20 years ago using time-domain THz spectroscopy. On the one hand, the authors claim that a “careful phase referencing” is not needed (like in THz electro-optic sampling). On the other hand, the stabilized phase shape of the THz pulses is also implicitly required by the reported method. Then, both methods actually rely on a precise measurement and control of the delay between two ultrashort optical pulses.

We agree with the referee that our time-of-flight method is similar to the more traditional time-domain THz spectroscopy in the sense that they both use two ultrafast laser pulses with a relative time delay. Yet, we would like to emphasize the following key advantages that make our method unique.

First, we directly sample the phase of the phonon polariton at the surface of the sample with time-resolved second harmonic generation. This is the only measurement needed to obtain the phase and amplitude of the phonon-polariton, making the subsequent data analysis very easy. In the traditional time-domain THz spectroscopy (TDS), phase sampling is performed in an external electro-optic (EO) sampling crystal. This external detection not only requires a more complex optical setup to refocus the transmitted THz beam but also distorts the phase of the THz pulse due to the crystal’s own dispersive properties¹. Therefore, a careful characterization of the EO sampling response of the EO sampling crystal is needed in order to extract the correct frequency dependent phase. This is usually done by measuring a reference THz pulse and typically involves phase unwrapping analysis to extract the phonon-polariton dispersion.

Second, our method offers the unique advantage to study the polariton dispersion of microscopic samples smaller than the diffraction limit of THz radiation ($\sim 300 \mu\text{m}$). This is because our near infrared probe can be focused to a spot with a few micrometer diameter. Therefore, our method can be used to study polaritons across domains and localized to domain boundaries²⁻⁴, as well as two-dimensional few layer flakes. This is impossible in conventional TDS.

Third, in TDS the sample is typically required to be either thick enough so that the internal reflections are well separated or thin enough so that the internal reflections almost overlap. In the intermediate range when the reflections having finite overlap, they substantially complicate the data analysis of TDS data and can lead to artifacts of the reconstructed phonon-polariton dispersion⁵. In our measurement, the sample thickness can be below $100 \mu\text{m}$ because the reflections are well-separated in the time-frequency space due to the simpler phase and amplitude extraction with our wavelet transform analysis. We demonstrate this advantage in our experiment

on MnPS₃ where the sample thickness is below 100 μm, leading to large overlap between back-reflected pulses.

To convey these three advantages better, we added a discussion paragraph (**Line 193-205**) and modified the wording in our manuscript accordingly.

The data analysis involves a wavelet transform which inevitably leads to uncertainties in time and frequency. It would be helpful if the authors commented on the accuracy and precision of the presented method, and ideally compared it with that achieved using time-domain THz spectroscopy.

We thank the reviewer for raising this important point. We have added a new supplementary section (**Supplementary Note 6**) dedicated to estimating the uncertainties of the wavelet transform analysis. We also provide the essential details of this analysis here.

The time-frequency resolution is related to the wavelet chosen for the wavelet transform. We chose the generalized Morse wavelet (GMW) with parameters $\beta = 120$, $\gamma = 3$ as the mother wavelet^{6,7}. As can be seen from Fig. R1 below, the time-frequency resolution varies with frequency. At the low frequency side of the phonon polariton (1.5 THz) the frequency resolution is 0.06 THz and the time resolution is 1.4 ps (Fig. R1a, red wavelets). The values are 0.15 THz and 0.5 ps at the high frequency side (4 THz) (Fig. R1a, blue wavelets). Here the resolution of the wavelet is defined by the square root of its variance.

On the other hand, the time steps taken in our measurement is 0.033 ps and the measurement time window of 20 ps allow a frequency resolution of 0.05 THz. Because of this oversampling, we can further improve the time/frequency resolution by fitting constant frequency/time line cuts with gaussian functions. We perform the gaussian fitting along constant frequency cuts for PP component smaller than 3.5 THz (Fig. R1b) and constant time cuts for larger than 3.5 THz (Fig. R1c). The uncertainty of this method is then the superposition of the experimental sampling step uncertainty and the uncertainty of the peak position obtained by the gaussian fitting. The final uncertainty window is shown by the solid filled region in Fig. R1d. The time uncertainty is found to be better than 0.05 ps and the frequency uncertainty better than 0.03 THz.

Fig. R1. Time and frequency resolution of the wavelet analysis. **a**, Wavelet transformed spectrogram aligned with representative wavelet. The real part of the time-domain wavelets is shown in the lower panel, and the frequency-domain wavelet in the left panel. **b**, Constant-frequency linecuts of the spectrogram and fitting with two gaussian peaks. The left (near zero delay) and the right peaks correspond to the incident pulse and the reflected pulse, respectively. **c**, Constant-time linecuts of the spectrogram and fitting with two gaussian peaks. The left (lower frequency) and the right peak correspond to the reflected PP and the localized phonon, respectively. **d**, Timing of the incident pulse and the reflected pulse. The light-shaded region shows the wavelet uncertainty window. The solid-shaded region shows our final time and frequency uncertainty.

In comparison, traditional time-domain THz spectroscopy has a similar time-resolution (and therefore a wavevector resolution) compared to our studies, bonded by the pulse width of the near-IR probe. The frequency resolution of TDTS is dependent on the sample thickness, since a back-reflected pulse will determine the available time window for performing Fourier transform. In the case of our NiI_2 sample, the back-reflected pulse arrives after 6 ps, which drastically limits the sampling window and leads to a frequency uncertainty of about 0.08 THz. For thinner samples, such as the MnPS_3 under study, the frequency resolution is expected to be worse due to the shorter sampling window. Therefore, our time-of-flight approach has comparable and potentially better time and frequency resolution compared to traditional TDTS.

The above results are now discussed in Supplementary Note 6 and also provided in main text **Line 145-148** and **155-157**.

The sample thickness must be specified (it is crucial not only for quantitative analysis of the dispersion relation, but also for the correct interpretation of the time-dependence of the probe signal).

We thank the reviewer for pointing out this oversight. We measured the sample thicknesses of the NiI_2 and MnPS_3 samples to be $(250 \pm 20) \mu\text{m}$ and $(80 \pm 20) \mu\text{m}$, respectively. These values are now specified in the **Methods** section. We used these values to update the experimentally obtained dispersion relation with an absolute measure of the wavevector in **Fig. 3** (also attached below in Fig. R2).

Fig. R2. Measured PP dispersion and Lorentz-model fitting with absolute measure of k . The error bars show the frequency and momentum uncertainties of our measurement at representative points.

The temperature of the measurement has to be specified in the manuscript to clarify which phase is actually studied.

We set a base temperature of 7 K and 12 K for the measurements on NiI_2 and MnPS_3 , respectively. At this temperature NiI_2 is in its multiferroic ground state and MnPS_3 forms a Néel type antiferromagnetic phase. We have added this information in the main text (**Line 86** and **Supplementary line 123**) and the temperature dependent results are also available in the Supplementary Note 9.

The work is certainly relevant and interesting, but I doubt that it is suitable for Nature Communications. I'm convinced that after appropriate revisions, the manuscript would be perfectly suitable for a journal like Phys. Rev. B. There are numerous interesting and important results, which are now hidden in the Supplementary Information while they would deserve publication in the body of a regular paper.

We thank the reviewer for their positive assessment of the relevance of our work. We believe that we have adequately responded to all the questions and comments and appreciate the recommendation to publish our work with these revisions. Due to its novelty, versatility and other above-mentioned advantages of our technique has over other conventional approaches, such as TDTs, we firmly believe that our work is an excellent fit for *Nature Communications* and its readership.

Questions and technical comments:

* Why magnetic materials and particularly van der Waals magnets were chosen for the study?
??

We want to point out that our method is applicable to a wide range of material systems and not limited to van der Waals materials. We choose to showcase our approach in two van der Waals magnets for a number of key reasons. First, compared to optical grade LiNbO_3 , where TDTs has

been used to study phonon polaritons, our samples are both imperfect crystals (see Fig. R3). This demonstrates that our method is suitable to measure the phonon-polariton dispersion of wide variety of crystals without the need for advanced sample preparation steps. Second, we chose a very thin piece of MnPS₃, so that we can demonstrate our capability to obtain polariton dispersion relation in very thin material systems, beyond the capability of TDTs. Here, the van der Waals nature of the samples was key to obtain optically thin samples without the need for sample polishing. These two points demonstrate the sample tolerance and general applicability of our time-of-flight approach. Finally, the macroscopic material properties in these materials are sensitive to the lattice degrees of freedom. In NiI₂, lattice and magnetic degrees of freedom are strongly coupled giving rise to type-II multiferroic ground state. Similarly, MnPS₃ belongs to a family of van der Waals magnets (MPX₃, M: Fe, Mn, Co; X: S, Se) where strong spin-lattice coupling and magnon-phonon hybridization has been observed⁸⁻¹⁰. Fingerprints of this spin-lattice coupling can appear in the phonon polariton dispersion (see Fig. 4 and Line 160-179). We have modified the corresponding sections in the main text to strengthen these points (Line 75-78).

Fig. R3. Optical image of the NiI₂ (a) and MnPS₃ (b) sample under study. The red dots on the sample depict the probe spot size used in the experiment.

* The explicit relation between $\omega(k)$ and $v_g(\omega)$ should be presented in Eq. (1)

We have changed Eq.1 accordingly to the explicit relation between $\omega(k)$ and $v_g(\omega)$.

* The beams hit the sample at 45 degrees while the simulation is performed in 1D. How was the oblique incidence reflected in the simulations and in the results?

The oblique incidence affects the simulations and the experimental results in three ways:

- 1) The momentum direction of the phonon polariton will not be strictly along the out-of-plane direction;
- 2) The optical distance of the phonon polariton inside the sample will be larger than two times of the sample thickness;
- 3) The phonon polariton wave packet will have a finite walk-off along the in-plane direction.

To understand how these three factors affect our results, we first provide an estimation of the angle between the phonon-polariton wavevector and the surface normal. The incidence angle of

the THz pump is 45° . This angle will be reduced inside the sample due to refraction at the surface. The refractive index of NiI2 in the relevant spectral range is greater than ~ 3 (see ref. 30). According to Snell's law, this yields an angle of about 14° between the polariton propagation direction and the surface normal. Therefore, the wavevector can be decomposed as $(0.97e_\perp + 0.24e_\parallel)k_0$, and we are probing the dispersion relation along a direction close to the out-of-plane direction. For point 2), the optical distance of the different frequency component of the phonon polariton is now given by $d = \frac{2l}{\cos\theta}$. With $\theta < 14^\circ$, d lies between $2l$ and $2.06l$, which only give rise to an uncertainty smaller than 3 % and therefore minimally affects the extraction of polariton dispersion relation. For point 3), the walk-off distance, given by $2l \tan\theta$, is estimated to be smaller than $130 \mu\text{m}$, which is within our THz beam spot size. Therefore, the reflected polariton wavepacket will not evade the second harmonic probe as long as the probe and pump center are spatially overlapping.

To supplement this semi-quantitative analysis, we re-performed the FDTD simulation with a 45° angle of incidence. The result is shown in Fig. R4. These simulations yield the same result and confirm that the oblique incidence has a negligible effect on our results. Yet, we agree with the reviewer that the measurement should ideally be performed in normal incidence geometry. We added discussions concerning these issues and the additional simulation in **Line 129-132, 149-150** as well as **Supplementary Note 5**.

Fig. R4. 1D simulation and realistic simulation and their comparison. **a**, the wavelet transform of the 1D-simulated time trace as shown in Fig. 3a. **b**, the wavelet transform of realistic simulation with a 45° angle of incidence. **c**, the arrival time of the incident pulse and the reflected pulse extracted from **a** and **b**. No significant difference is observed.

* Many references are incomplete (mostly a page/article number is missing) [10, 17, 23, 24, 27, ...].

We thank the reviewer for the careful reading. We have added the missing page numbers and carried out additional proofreading to the entire manuscript.

* The spheres in Fig. 1a look grey rather than purple as written in the caption

We have adjusted the color to be consistent with the caption.

* Line 86 the authors claim that "... we can rule out a replica of our pump or probe pulses (see Fig. 1d)". Why the replicas are ruled out?

This statement is based on the EO sampling trace shown in Fig. 1d of the main text. The EO sampling time trace does not show a replica within 10 ps after the initial pump pulse. Therefore, the signal revival that we observe in NiI_2 after 6 ps is necessarily coming from the sample and not from the pump or probe pulses. We also clarified this point in the manuscript (**Line 97-99**).

* The discussion in the paragraph at lines 89 - 100 could be improved; it is now difficult to decode the main message.

We have rephrased the paragraph and hope to have increased its clarity.

* Would it be possible to observe also the upper polariton branch using the developed approach?

The time-of-flight method developed in our work is applicable to both lower and upper polariton branches. Any polariton branch that falls in our THz bandwidth can be detected and measured. In our work on MnPS_3 (see Supplementary Note 10), a portion of the upper polariton branch is detected. In NiI_2 , however, the upper polariton branch does not fall in our THz bandwidth and is therefore not detected. To support our conclusion, we carried out additional simulation with an artificially increased THz pump bandwidth. We used the second derivative of a gaussian peak (Fig. R5a) to create a large bandwidth (Fig. R5b) THz pulse. Upon sending this pulse to the sample, both the lower and upper polariton branches are clearly visible in the wavelet transform (Fig. R5c).

We have added one sentence in the main text (**Line 184-185**) as well as **Supplementary Note 10** to clarify this point.

Fig. R5. Simulation of time-of-flight measurement with a large bandwidth THz pulse. a, Time-trace of the THz pulse used for the simulation. **b,** The spectrum of the THz pulse in **a**. **c,** Simulated spectrogram of the time-of-flight measurement with the THz pulse in **a**. Two branches of time-dependent frequency peaks are visible above and below the TO frequency, corresponding to the upper and the lower polariton branch.

Response to Reviewer #2

2023. 7. 19
Review Report

Manuscript: #NCOMMS-23-27251

Manuscript Title: "Time-of-flight detection of terahertz phonon polariton", by Tianchuang Luo et al.

The authors propose a new approach based on a time-of-flight measurement to determine the phonon-polariton dispersion of van der Waals magnets NiI₂ and MnPS₃. Although their proposed technique is basically interesting, questions are raised after reading the paper as stated below.

We thank the reviewer for their carefully reading and accurate summary of the main findings of our manuscript and appreciate their positive remarks. In the following we will address their questions and concerns point-by-point.

(1) The authors said, "Determining the polariton dispersion relation is a highly nontrivial task, and traditional methods such as Raman scattering and transient grating spectroscopy require careful tuning of the incidence angle of the photon to determine the wavevector of the polariton". This is actually true. However, the conventional Raman scattering method tracks Raman active modes, while the authors' proposed technique can access the IR active modes. Therefore, these might be complementary to each other. Even utilizing the authors' technique, I guess, the phonon-polariton (PP) dispersion of Raman active modes cannot be obtained.

The reviewer's remark is generally correct if the crystal lattice preserves inversion symmetry. In these materials, one can strictly distinguish between *gerade* (Raman) and *ungerade* (infrared active) phonon modes. Importantly, phonon-polaritons (PPs) form due to strong coupling between electromagnetic waves and transverse optical (TO) phonons with dipole moment, i.e., infrared (IR) active modes. Therefore, in a material with inversion symmetry, Raman phonons do not hybridize with light. To our knowledge, every study that used Raman scattering to measure the PP dispersion were performed in inversion symmetry broken crystals (such as LiNbO₃) or detecting surface-phonon-polaritons at the interfaces with intrinsically broken inversion symmetry. In such cases all the IR modes are Raman-active too and this is why PPs can be observed in a Raman measurement. On the other hand, in centrosymmetric systems Raman modes do not form PPs and Raman scattering cannot be used to measure the PP dispersion. Instead, other techniques such as hyper-Raman scattering¹¹ or quadrupole-assisted four-wave-mixing¹² were used to detect PPs in systems with inversion symmetry. Such examples are, however, rare.

In short, the phonon-polaritons studied by conventional Raman experiments are simultaneously IR-active and can therefore also be probed by the method developed in our work. In addition, we have the potential capability to study phonon-polaritons in inversion symmetric systems to which conventional Raman technique is not sensitive.

To clarify these symmetry restrictions on the detection of PPs in the Raman and IR spectroscopies, we added a sentence at **Line 47-49** to discuss these symmetry selection rules.

(2) To reveal not only the PP dispersion but also the anharmonicity of the PP is crucial for understanding the underlying physics of quantum materials and advancing THz technologies. In previous papers [Refs. 18, 22, 23, 24 and JAP 123, 174103, 2018], anharmonic phonon-polariton characteristics such as damping rate, coupling and so on, are observed. However, the experimental results in this paper seems within a linear regime. Does the authors' technique evaluate the anharmonicity of the PP? If not, the proposed technique is not powerful so much.

We appreciate the reviewer for raising this important point. In the papers that the reviewer suggested, anharmonic phononic effects, including coupling of phonon to other optical and acoustic phonon modes as well as relaxational modes, manifest themselves as the frequency and temperature dependence of the damping rate. As will be demonstrated below, our method is fully capable of providing such information about PPs, which will provide access to PP anharmonicities.

Motivated by the reviewer's suggestions, we demonstrate here anharmonic coupling of the PP in NiI_2 to other collective modes. Here, we determine the frequency-dependent damping rate by comparing the incident amplitude (Fig. R6a, blue line) and the back reflected amplitude (Fig. R6b red line) in the spectrogram, which are related by a ratio of $e^{-2\alpha l} r(1+r)S$, where α is the damping rate and is related to the imaginary part of the complex PP wavevector ($\tilde{k} = k + i\kappa$) $\kappa = \alpha$. r is the reflection coefficient at the boundary of the sample. The factor $(1+r)$ accounts for the fact that the reflected pulse and its further reflection at the front surface are detected simultaneously. S is a constant correction factor that accounts for the slight spatial offset between the incident and the reflected pulse. In Fig. R6b we plot the relation between PP frequency ω and κ (dark blue line). κ increases when the frequency approaches the transverse phonon mode as expected from the Lorentz model. Interestingly, we observe a shoulder peak at around 3.1 THz (dashed gray line). This anomalous damping rate can be explained by anharmonic coupling to another collective mode (potentially a magnon mode) at the shoulder peak frequency¹³. The imaginary and the real part of the dispersion relation (Fig. R6c) can be simultaneously described by the bilinear coupling between PP and another mode, with the coupling parameters extracted as $\frac{b_{12}b_{21}}{4\pi^2} = (1.6 \pm 0.1) \text{ THz}^2$, which is around 3 % to the oscillator strength of the phonon polariton $\frac{b_{1E}b_{E1}}{4\pi^2\epsilon_0} = (61 \pm 2) \text{ THz}^2$.

Fig. R6. Real and imaginary part of the PP dispersion and coupled oscillator fitting. **a**, the amplitude of the incident and reflected pulse obtained from Gaussian fitting of the spectrogram. **b**, the imaginary part of the dispersion (dark blue line) obtained from **a** and fitting with coupled oscillator model (light blue line). **c**, the real part of the dispersion obtained from time-of-flight information (dark red line), fitting with Lorentz model without damping (violet line), and fitting with coupled oscillator model, the same model used for the imaginary part fitting (orange line).

To confirm that this anomaly is not caused by the oblique incidence of our THz pump pulse, we repeated our FDTD in a configuration more closely mirroring our experiment (Fig. R7b), where we simulate a focused THz beam source with a beam width comparable to our experiment and an angle of incidence of 45° . Only a single phonon is considered in the simulation. On applying the same extraction method as in Fig. R6b, the simulated phonon polariton damping rate does not show any anomalies close to 3.1 THz (Fig. R7c). Therefore, we can conclude that our time-

of-flight scheme can also decode the anharmonic couplings. In future experiments, the extraction of damping rate can be made more accurate by using a plane-wave, normal-incident THz pulse.

We have added above discussions to **Line 160-179** together with **Fig. 4** and **Supplementary Note 5** and **7** to demonstrate the capability of understanding anharmonicity of our time-of-flight approach. We also added **ref. 35** to provide the readers with more background information.

Fig. R7. Comparison between 1D and realistic simulation. **a**, the wavelet transform of the time trace in 1D simulation (also in Fig. 3a of the manuscript). **b**, the wavelet transform for realistic simulation. **c**, Comparison of the PP damping rate extracted from **a** and **b**. **d**, Comparison of the arrival time of the incident and reflected pulse extracted from **a** and **b**.

(3) The dispersion of phonons and magnons is generally obtained by using neutron scattering method, which may give us the dispersions in the whole Brillouin zone. In contrast, optical measurements only tracks the dispersion around the gamma point. Could the authors compare the measured PP dispersion of NiI2 and MnPS3 with the data obtained by neutron scattering?

As the reviewer correctly points out, our measurement only allows us to measure the dispersion of the polaritons close to the gamma point. Yet, before we answer the reviewer's question, we want to clarify that the dispersion measured by our technique and the one measured by neutron diffraction are complementary. As pointed out by the reviewer, neutron scattering is most suitable for studying the dispersion of phonons/magnons across the entire Brillouin zone ($\sim 1 \text{ nm}^{-1}$). In most part of the Brillouin zone, the energies of photons and phonons are strongly mismatched. As a result, polaritons only form close to the gamma point with a wavevector of the order of $0.1 \mu\text{m}^{-1}$ where the energy and momenta are matched and the bare photon and phonon dispersions cross each other. Such a small momentum is inaccessible to standard inelastic

neutron scattering setups¹⁴. On the other hand, our technique is sensitive to momenta close to the Brillouin zone center and measures the coupling of light to IR-active phonon modes and the dispersion of the resulting polariton. Finally, our measurement also reveals the bare phonon frequency, which can in principle be compared to results obtained with inelastic neutron scattering extrapolated to zero momentum. However, the neutron scattering data on zone-center phonon modes in $\text{NiI}_2/\text{MnPS}_3$ are not available to our knowledge.

Following the reviewer's comment, we have added one sentence of discussion at **Line 44-46**.

(4) The authors should state the experimental condition more clearly.
What is the sample thickness?

We thank the reviewer for pointing out these crucial oversights. The NiI_2 and MnPS_3 samples we used in our experiment are (250 ± 20) μm and (80 ± 20) μm thick, respectively. These values are now specified in the **Methods** section. Armed with this information, we have also updated the experimentally obtained dispersion relation with the absolute values of wavevector in **Fig. 3** and **4**.

To implement the authors' time-of-flight measurement, the dataset of each temporal window (Window A, B and C in Fig. 2) is separately needed, implying that overlapping of the dataset among the windows might be undesirable.

The Fourier transform with varying temporal window in manuscript Fig. 2 serves to convey the message of the time dependent spectral content of the phonon polariton field. For the time-of-flight analysis shown in manuscript Fig. 3, we used a wavelet transform. The wavelet transform performs an inner product between the time trace with a scaled wavelet, which can be continuously translated along the time axis without the use of a time window. We found that the wavelet transform better captures the polariton feature compared to the short time Fourier transform (STFT) with windowing, as compared in Fig. R8. A 4 ps time window produces the best STFT result, yet it fails to capture the time-frequency relation close to the signal revival at around 6 ps and is susceptible to interference fringes. Therefore, the time-frequency analysis in our main text is carried out with wavelet transform where no temporal window is needed.

To better delivery this message, we carefully describe the details of the wavelet transform in **Methods** and introduced additional clarification in the main text at **Line 117-119** and **Supplementary Note 3** to help the readers better understand our approach.

Fig. R8. Comparison between wavelet transform and STFT. a, the wavelet transform of the experimental time trace. b, the STFT of the experimental time trace with a 4 ps time window.

In addition, as seen in the inset of Fig. 2a and Fig. 1S, the incident angle of THz and probe pulses is tilted against the sample, implying that the optical distance is not the sample thickness.

The reviewer is correct that our THz pump and optical probe beams do not hit the sample at normal incidence. To understand how this condition affects our results, we first estimate the angle between the phonon-polariton wavevector and the surface normal. The incidence angle of the THz pump is 45° . This angle will be reduced inside the sample due to refraction at the surface. The refractive index of NiI_2 in the relevant spectral range is greater than ~ 3 (see ref. 30). According to Snell's law, this yields an angle of about 14° between the polariton propagation direction and the surface normal. The optical distance of the different frequency components of the phonon polariton is now given by $d = \frac{2l}{\cos\theta}$. With $\theta < 14^\circ$, d lies between $2l$ and $2.06l$, which only give rise to an uncertainty smaller than 3% and therefore minimally affects the extraction of polariton dispersion relation.

To supplement this semi-quantitative analysis, we re-performed the FDTD simulation with 45° angle of incidence. The result is shown in Fig. R4. These simulations yield the same result and confirm that the oblique incidence has a negligible effect on our results. Yet, we agree with the reviewer that the measurement should ideally be performed in the normal incidence geometry. We added a discussion concerning these issues (**Line 129-132 and 149-150**) and added the additional simulations to the **Supplementary Note 5**.

Moreover, I guess, the authors need information on the refractive index of the probe.

We agree with the reviewer that the refractive index of the probe is important for the determination of phase matching conditions. In our case we choose a specific limiting case where the absorption coefficient of the second harmonic of the NIR probe is very large (see ref. 32-33). Because of this, we are only detecting SH light from a thin layer with a thickness smaller than $\sim 10 \mu\text{m}$ below the surface. Therefore, the probed region is much smaller than the wavelength of the THz phonon polariton, rendering phase-matching conditions irrelevant. We also note that this requirement, although sounds restrictive, is not difficult to satisfy. In general, such a condition can be fulfilled by choosing a probe wavelength where the SH is larger than the electronic bandgap of the material under study. We also provided extra discussions in the main text to improve clarity (**Line 100-102**).

The regions of the dispersion are limited around $10\sim 35k(\text{pai}/l)$ in Fig. 3 and $3\sim 17k(\text{pai}/l)$ in Fig. S5. What kind of the experimental condition contributes to the limitation of the observable region? Could the authors change the region of the dispersion? In case of the previous works using the phase-match conditions [23, 24, JAP 123, 174103, 2018], the dispersion region can be changed by changing the pump wavelength.

We thank the reviewer for asking this question. The accessible range of our time-of-flight technique is indeed an important parameter. Two factors limit the available dispersion range in our method. The first and the most important parameter is the THz bandwidth (half maximum at around 1-5 THz). The bandwidth limits the frequency window where the PP can be excited, and therefore also limits the observable PP wavevector range. This limitation can be lifted by using different THz sources such as optical rectification in different nonlinear media¹. The second limitation is given by the damping rate of the PP. When the damping rate of PP is too large for it to travel through the sample, our method will not be able to pick up the phonon polariton signal.

Considering these two limitations, the frequency window in NiI_2 is limited to between 1.5 THz - 4 THz.

In the previous works pointed out by the referee, the dispersion relation is measured through phase-matching of the probe beam to the phonon-polariton. In this case, the authors achieved phase-matching by changing the laser wavelength. This allowed them to access discrete points in the dispersion relation each corresponding to a different laser wavelength. To better illustrate the advantages of our method, we estimate the corresponding wavelength-range to achieve phase-matching in the same frequency range as we report here. According to infrared spectroscopy (ref. 30) and linear dichroism measurements (ref. 28), we can estimate a refractive index of 2.4 and a refractive index anisotropy of around 0.018. To cover our frequency window (1.5 THz - 4 THz), one would need to tune the pump wavelength either between 600 nm and 2900 nm for forward scattering or 600 nm and 200 nm for backward scattering. Both ranges are difficult to access with a single laser setup using conventional optical parametric amplifiers and associated systems. Although a combination of forward and backward scattering will potentially yield the same frequency window, an extensive set of at least 30 scans at different wavelengths is required to achieve the same frequency resolution as in our experiment. Therefore, we believe that, compared to phase-matching techniques based on wavelength tuning, our method is superior. It allows us to measure the full dispersion without changing the THz pump wavelength and we obtain the continuous dispersion rather than discrete points. The frequency and wavevector window can potentially be further extended by using different THz sources¹. We have also added this discussion to **Line 193-198**.

The authors utilized the wavelet analysis. How much does the frequency resolution coming from the analysis?

We thank the reviewer for raising this important point. We have added a new supplementary section (**Supplementary Note 6**) dedicated to estimating the uncertainties of the wavelet transform analysis. We also provide the essential details of this analysis here.

The time-frequency resolution is related to the wavelet chosen for the wavelet transform. We chose the generalized Morse wavelet (GMW) with parameters $\beta = 120$, $\gamma = 3$ as the mother wavelet^{6,7}. As can be seen from Fig. R1 below, the time-frequency resolution varies with frequency. At the low frequency side of the phonon polariton (1.5 THz) the frequency resolution is 0.06 THz and the time resolution is 1.4 ps (Fig. R1a, red wavelets). The values are 0.15 THz and 0.5 ps at the high frequency side (4 THz) (Fig. R1a, blue wavelets).

On the other hand, the time steps taken in our measurement is 0.033 ps and the measurement time window of 20 ps allow a frequency resolution of 0.05 THz. Because of this oversampling, we can further improve the time/frequency resolution by fitting constant frequency/time line cuts with gaussian functions. We perform the gaussian fitting along constant frequency cuts for PP component smaller than 3.5 THz (Fig. R1b) and constant time cuts for larger than 3.5 THz (Fig. R1c). The uncertainty of this method is then the superposition of the experimental sampling step uncertainty and the uncertainty of the peak position obtained by the gaussian fitting. The final uncertainty window is shown by the solid filled region in Fig. R1d. The time resolution is found to be better than 0.05 ps and the frequency resolution better than 0.03 THz. This result is also provided in main text **Line 145-148** and **155-157**.

Is the sample the uniaxial crystal or isotropic one? The polarization of THz and probe pulses to the crystal axis is probably crucial for the analysis.

Both NiI_2 and MnPS_3 are optically anisotropic crystals, where the out-of-plane direction (*c*-axis) is significantly distinct from the in-plane directions (*ab*-axis). Therefore, the referee is correct in their assessment that the polarization of the THz and probe pulses with respect to the crystal axis matters.

In our experiment, the pump pulses are *s*-polarized with respect to the sample surface (the electric field is parallel to the sample surface), therefore even for oblique incidence as in our experiment, we only probe the in-plane electrodynamics. The in-plane optical anisotropy is small in both NiI_2 (see ref. 28) and MnPS_3 ¹⁵. This provides us with a clear measurement of the dispersion of the phonon-polaritons that are polarized along the *ab*-directions of the crystals.

For the probe pulse, we choose a polarization where we have a significant amount of second harmonic generation, enabling us to be sensitive to the inversion symmetry breaking induced by the PP.

Following the reviewer's suggestion, we have included this information now in the **Methods section (Line 225)**.

Overall, the paper is well written and basically interesting. However, the authors should claim more clearly the technical merit of their experimental technique or pros and cons to the previous works.

We appreciate the reviewer's positive remarks and hope the changes we made to the manuscript and our responses to their concerns now convince them on the importance and the merit of our work.

Response to Reviewer #3

The authors report a method to measure phonon polariton dispersion curves with a time of flight method. The front surface of a sample is excited with a broadband THz pulse and phonon polaritons are detected after propagation through the sample.

The technique is interesting and could be useful, but in its current form, the manuscript reads as a brief introduction to a new technique and analysis method. The impact of this manuscript would be greater (and better suited for Nature Communications) with the additional information:

1) Besides a measurement of a portion of the phonon-polariton dispersion curve, some discussion of what this means and why it is important for the studied materials is needed. Why are NiI_2 and MnPS_3 potentially useful and what information about the materials and their usefulness does the dispersion curve give us? They introduce the interest in polaritons generally in the intro, but then never tie it together with the experimental results.

We agree with the referee that in the current version of the manuscript the discussion of the specific material choice, in our case NiI_2 and MnPS_3 , is not sufficiently stressed. Yet, we want to emphasize that the primary focus of our manuscript is the demonstration of a new technique and the specific choice of material is only of secondary interest.

However, both NiI_2 and MnPS_3 are interesting systems in their own rights. They display strongly coupled degrees of freedom, with NiI_2 being a multiferroic material, and the MPX_3 ($M = \text{Fe, Ni, Mn, Co}$; $X = \text{S, Se}$) class of materials are known for their strong spin-lattice coupling⁸⁻¹⁰. The manifestations of these couplings can be observed in the phonon-polariton dispersions, as we have now demonstrated also in our response to reviewer 2.

Generally, the real part of the phonon polariton dispersions quantify the coupling strength between light and a specific phonon modes (the ω_L or $b_{1E}b_{E1}$ parameter). Such information can be used to compare the light matter coupling strength in different materials or different modes. For example, the extracted light-matter coupling in NiI_2 is $\frac{b_{1E}b_{E1}}{4\pi^2\epsilon_0} = (61 \pm 2) \text{ THz}^2$ and for MnPS_3 the value is $(50.6 \pm 0.4) \text{ THz}^2$. Therefore, we can conclude that the NiI_2 couples stronger to in-plane polarized light in this frequency range (1-5 THz).

In addition, anomalies in the dispersion curve can highlight anharmonic coupling to other degrees of freedom, which is now also demonstrated in our work in **Fig. 4** and **Line 160-179**. Understanding the anharmonic couplings helps us to identify effective pathways for the dynamical control of material properties, potentially using a stronger THz pulse.

Beyond the linear response discussed in the main text, our technique would also allow us to measure the phonon-polariton dispersion in the nonlinear regime (see ref. 18). Unfortunately, both NiI_2 and MnPS_3 seem to only have weak nonlinearities, which remain hidden at the high field strength (350 kV/cm) used in our experiments.

To sum up, measurement of the phonon polariton dispersion is a natural extension to Raman scattering and other spectroscopic techniques and provides complimentary information about the dynamics of solid matter. Following the referee's advice, we added

- 1) discussions on the general implications of PP dispersion relation (**Line 37-43, Line 204-205**),
- 2) discussions about what our data implies for the specific cases of NiI_2 (**Line 75-78 and 160-179** together with **Fig. 4**),
- 3) a comparison between NiI_2 and MnPS_3 (**Line 190-192**).

Hopefully these changes will convey the importance of understanding PP dispersion.

2) What are the limitations of this experimental method? For what phonon-polariton attenuation coefficient versus the sample thickness is the experiment still possible? (The PP needs to be large enough amplitude to still be detected after propagating across the sample)

We agree that our technique does not work with large attenuation coefficients (similar to TDTS). One can estimate the maximum measurable attenuation coefficient (α) in the following way. In order to achieve a few picosecond separation between the initial time-zero and the arrival of the first reflection, the sample should not be substantially thinner than 100 μm . In turn, to obtain a reliable signal the reflection should not be too much smaller than 10% of the initial THz pulse, i.e., $\frac{A_{\text{reflected}}}{A_{\text{incident}}} = e^{-100 \mu\text{m} \alpha} > 10\%$. This means that our measurement can yield reliable results for attenuation coefficients of about 100 cm^{-1} . This limitation is not too stringent as this degree of attenuation can only be reached close to collective mode resonances where the dispersion is negligible anyways.

We have added a discussion regarding this limitation to the main text at **Line 193-197**.

3) Typically, PP dispersion curves are plotted as two parts - the dispersive part and the absorptive part. Can you extract the absorptive part from this data? If so, please include it in the plots.

We thank the reviewer for this comment. Our time-of-flight approach is also capable of extracting the absorptive part of the dispersion. The frequency-dependent damping rate is obtained by comparing the incident amplitude (Fig. R6a, blue line) and the back reflected amplitude (Fig. R6b red line) in the spectrogram, which are related by a ratio of $e^{-2\alpha l} r(1+r)S$, where α is the damping rate and is related to the imaginary part of the complex PP wavevector ($\tilde{k} = k + i\kappa$) $\kappa = \alpha$. r is the reflection coefficient at the boundary of the sample. The factor $(1+r)$ accounts for the fact that the reflected pulse and its further reflection at the front surface are detected simultaneously. S is a constant correction factor that accounts for the slight spatial offset between the incident and the reflected pulse. In Fig. R6b we plot the relation between PP frequency ω and κ (dark blue line). κ increases when the frequency approaches the transverse phonon mode as expected from the Lorentz model. Interestingly, we observe a shoulder peak at around 3.1 THz (dashed gray line). This anomalous damping rate can be explained by anharmonic coupling to another collective mode (potentially a magnon mode) at the shoulder peak frequency¹³. The imaginary and the real part of the dispersion relation (Fig. R6c) can be simultaneously described by the bilinear coupling between PP and another mode, with the coupling parameters extracted as $\frac{b_{12}b_{21}}{4\pi^2} = (1.6 \pm 0.1) \text{ THz}^2$, which is around 3 % to the oscillator strength of the phonon polariton $\frac{b_{1E}b_{E1}}{4\pi^2\epsilon_0} = (61 \pm 2) \text{ THz}^2$.

The absorptive part of the dispersion is now provided in **Fig. 4** and discussed at **Line 160-179**.

4) With the emphasis on a time-of-flight type measurement, I naturally want to know for what situations would you need/want to probe from the back side of the sample? And what would be considerations for doing the measurement like that?

From our understanding, probing from the back and from the front of the sample will yield equivalent results and each have their own advantages. For samples with a large THz attenuation coefficient, probing from the back of the sample is beneficial since the THz phonon-polariton will only need to transverse the sample thickness once. Yet, we would lose crucial information about the incoming THz field, which naturally appears in our signal when we probe from the front. This is the main advantage of probing from the front. In addition, probing from the front leads to a longer interaction length inside the sample (a factor of two), which could lead to a higher sensitivity when the PP dispersion relation is weak. A more practical reason to probe the front is an easier experimental setup and alignment. Yet, the reviewer's comment is definitely thought-provoking, and it would be interesting to simultaneously probe both from the front and the back. This way, we can better understand the absorptive part of the dispersion by comparing the phonon polariton amplitudes at opposite sides of the sample.

We have also included this discussion in the manuscript at **Line 157-159** to enlighten the readers with potential future development of the time-of-flight method.

5) I see fitted values for the Lorentz model fitting of the PP dispersion relation for Nil_2, but they are absent from the SI for MnPS_3

We thank the reviewer for pointing out this important oversight. The PP parameters in MnPS₃ is $\omega_T = 4.78 \text{ THz}$ and $\omega_L = 5.04 \text{ THz}$. We have also added this information in the **Supplementary Note 11**.

Minor things that need addressing:

A) The text, letters, in the figures I think is too small

We have increased the font size in the figures as suggested.

B) Do we have error bars on the values for the measured PP dispersion? For a proof of principle measurement like this, introducing the technique along with it's limitations is important.

We have added error bars to the real and imaginary part of the dispersion in **Fig. 3** and **Fig. 4**. We thank the reviewer for addressing this oversight.

References

1. Seifert, T. *et al.* Efficient metallic spintronic emitters of ultrabroadband terahertz radiation. *Nat. Photonics* **10**, 483–488 (2016).
2. Wu, X. *et al.* Low-energy structural dynamics of ferroelectric domain walls in hexagonal rare-earth manganites. *Sci. Adv.* **3**, e1602371 (2017).
3. Nikitin, A. Y. *et al.* Real-space mapping of tailored sheet and edge plasmons in graphene nanoresonators. *Nat. Photonics* **10**, 239–243 (2016).
4. Rizzo, D. J. *et al.* Charge-Transfer Plasmon Polaritons at Graphene/ α -RuCl₃ Interfaces. *Nano Lett.* **20**, 8438–8445 (2020).
5. Duvillaret, L., Garet, F. & Coutaz, J. L. A reliable method for extraction of material parameters in terahertz time-domain spectroscopy. *IEEE J. Sel. Top. Quantum Electron.* **2**, 739–745 (1996).
6. Olhede, S. C. & Walden, A. T. Generalized Morse wavelets. *IEEE Trans. Signal Process.* **50**, 2661–2670 (2002).
7. Lilly, J. M. & Olhede, S. C. Generalized morse wavelets as a superfamily of analytic wavelets. *IEEE Trans. Signal Process.* **60**, 6036–6041 (2012).
8. Godejohann, F. *et al.* Magnon polaron formed by selectively coupled coherent magnon and phonon modes of a surface patterned ferromagnet. *Phys. Rev. B* **102**, 144438 (2020).
9. Mai, T. T. *et al.* Magnon-phonon hybridization in 2D antiferromagnet MnPSe₃. *Sci. Adv.* **7**, 3106–3135 (2021).
10. Vaclavkova, D. *et al.* Magnon polarons in the van der Waals antiferromagnet FePS₃. *Phys. Rev. B* **104**, 134437 (2021).
11. Inoue, K., Asai, N. & Sameshima, T. Observation of the Phonon Polariton in the Centrosymmetric Crystal of SrTiO₃ by Hyper-Raman Scattering. *J. Phys. Soc. Japan* **48**, 1787–1788 (1980).

12. Vianna, S. S. & De Araujo, C. B. Coherent excitation of phonon polaritons in a centrosymmetric crystal. *Phys. Rev. Lett.* **56**, 1475–1477 (1986).
13. Wiederrecht, G. P. *et al.* Explanation of anomalous polariton dynamics in LiTaO₃. *Phys. Rev. B* **51**, 916–931 (1995).
14. Ghose, S. Inelastic neutron scattering. in *Spectroscopic Methods in Mineralogy and Geology* 161–192 (2019). doi:10.1088/978-1-64327-114-9ch2.
15. Zhang, Q. *et al.* Observation of Giant Optical Linear Dichroism in a Zigzag Antiferromagnet FePS₃. *Nano Lett.* **21**, 6938–6945 (2021).

REVIEWERS' COMMENTS

Reviewer #1 (Remarks to the Author):

The authors provided convincing arguments that their method of characterization of phonon polaritons is a significant development. All my remarks (and I believe that also those of the other Referees) were addressed really thoroughly; the additional material is very helpful to understand the potential as well as the limitations of the method. I highly appreciate that the authors further exploited their method, e.g. by through the measurement of the imaginary part of the dispersion. Now I fervently recommend the manuscript for publication.

Reviewer #2 (Remarks to the Author):

Review Report

Manuscript: #NCOMMS-23-27251A

Manuscript Title: "Time-of-flight detection of terahertz phonon polariton", by Tianchuang Luo et al.

The authors propose a new approach based on a time-of-flight measurement to determine the phonon-polariton dispersion of van der Waals magnets NiI₂ and MnPS₃. In the revised manuscript, the authors reasonably address all the concerns raised by the reviewer(s). Especially, I was very impressed since not only the authors clearly stated the advantages of their approach in the revised manuscript but also performed further experiment/analysis to extract a bi-linear coupling between the Eu mode and another mode in NiI₂, showing the potential signature of their method to unveil the light-matter interaction of novel materials including van der Waals magnets. Hence, I recommend the paper is suitable for Nature Communications.

Optional

The authors introduce a correlation factor S into the eq. (3) probably due to resolving the mismatch between the actual experimental result and the simulation. It would be helpful for readers if the value of S is written somewhere in the text although this is not significant parameter.

Nil2 Reviewer Report

Response to Reviewer #1

The authors provided convincing arguments that their method of characterization of phonon polaritons is a significant development. All my remarks (and I believe that also those of the other Referees) were addressed really thoroughly; the additional material is very helpful to understand the potential as well as the limitations of the method. I highly appreciate that the authors further exploited their method, e.g. by through the measurement of the imaginary part of the dispersion. Now I fervently recommend the manuscript for publication.

We thank the reviewer for careful reading of the additional material and acknowledging the significance of our work. We hope the publication of this work will lead to deeper understanding of polaritons and, more generally, strong light-matter interaction physics.

Response to Reviewer #2

The authors propose a new approach based on a time-of-flight measurement to determine the phonon-polariton dispersion of van der Waals magnets Nil2 and MnPS3. In the revised manuscript, the authors reasonably address all the concerns raised by the reviewer(s). Especially, I was very impressed since not only the authors clearly stated the advantages of their approach in the revised manuscript but also performed further experiment/analysis to extract a bi-linear coupling between the Eu mode and another mode in Nil2, showing the potential signature of their method to unveil the light-matter interaction of novel materials including van der Waals magnets. Hence, I recommend the paper is suitable for Nature Communications.

We thank the reviewer for his/her appreciation of the revised manuscript, especially the extracted bi-linear mode coupling. We hope the publication of this work will lead to more studies on the light-matter interaction in novel van der Waals material.

Optional

The authors introduce a correlation factor S into the eq. (3) probably due to resolving the mismatch between the actual experimental result and the simulation. It would be helpful for readers if the value of S is written somewhere in the text although this is not significant parameter.

We thank the reviewer for pointing this out. We now provide the value of S (1.28 ± 0.01) in the Methods to provide readers with a complete picture while not interrupting the main message in the main text.